

# Identification of HMMR as a prognostic biomarker for patients with lung adenocarcinoma *via* integrated bioinformatics analysis

Zhaodong Li[1,*], Hongtian Fei[2,*], Siyu Lei[1], Fengtong Hao[1], Lijie Yang[1], Wanze Li[1], Laney Zhang[3] and Rui Fei[1,4]

[1] Department of Cell Biology, College of Basic Medical Sciences, Jilin University, Changchun, Jilin, China
[2] Department of Pharmacology, College of Basic Medical Sciences, Jilin University, Changchun, Jilin, China
[3] The College of Arts and Sciences, Cornell University, New York, USA
[4] Key Laboratory of Lymphatic Surgery Jilin Province, Jilin University, Changchun, Jilin, China
* These authors contributed equally to this work.

Corresponding author
Rui Fei, feirui@jlu.edu.cn

## ABSTRACT

**Background:** Lung adenocarcinoma (LUAD) is the most prevalent tumor in lung carcinoma cases and threatens human life seriously worldwide. Here we attempt to identify a prognostic biomarker and potential therapeutic target for LUAD patients.
**Methods:** Differentially expressed genes (DEGs) shared by GSE18842, GSE75037, GSE101929 and GSE19188 profiles were determined and used for protein-protein interaction analysis, enrichment analysis and clinical correlation analysis to search for the core gene, whose expression was further validated in multiple databases and LUAD cells (A549 and PC-9) by quantitative real-time PCR (qRT-PCR) and western blot analyses. Its prognostic value was estimated using the Kaplan-Meier method, meta-analysis and Cox regression analysis based on the Cancer Genome Atlas (TCGA) dataset and co-expression analysis was conducted using the Oncomine database. Gene Set Enrichment Analysis (GSEA) was performed to illuminate the potential functions of the core gene.
**Results:** A total of 115 shared DEGs were found, of which 24 DEGs were identified as candidate hub genes with potential functions associated with cell cycle and *FOXM1* transcription factor network. Among these candidates, *HMMR* was identified as the core gene, which was highly expressed in LUAD as verified by multiple datasets and cell samples. Besides, high *HMMR* expression was found to independently predict poor survival in patients with LUAD. Co-expression analysis showed that *HMMR* was closely related to *FOXM1* and was mainly involved in cell cycle as suggested by GSEA.
**Conclusion:** *HMMR* might be served as an independent prognostic biomarker for LUAD patients, which needs further validation in subsequent studies.

## INTRODUCTION

Lung cancer remains the most common cause of cancer-related death worldwide. More than 1,600,000 patients are newly diagnosed with lung cancer each year, which reduces patients' life quality and brings heavy financial burden to the patients (*Qu et al., 2020*). Lung adenocarcinoma (LUAD) is the most frequent histological type among lung cancers, which occupies approximately 40% (*Bai et al., 2019a*). Although the clinical strategies treating LUAD, such as chemotherapy, radiotherapy, targeted therapy, surgery, and immunotherapy, have been developed in recent decades, the 5-year survival rate of LUAD remains unsatisfactory, mainly because of the lack of effective prognostic biomarkers and therapeutic targets involved in the development and progression of the lung carcinoma (*Chen et al., 2019a*). It is therefore crucial to perform an integrated study to identify the core genes and comprehensively understand its relevant signaling pathways during the LUAD occurrence and progression.

Nowadays, the combination of genomics technology and bioinformatics analysis facilitated the wide application of gene expression profiles in a range of human cancers, providing a new insight into screening tumor-associated genes and identifying the core prognosis factors (*Kaushik et al., 2020*; *Huang et al., 2019*; *Liu et al., 2019*). Meanwhile, high-throughput technologies simplified the procedure of gene expression profiling and a growing body of expression data can be retrieved from public databases, which allows us to further study the underlying molecular mechanisms and medicine targets in LUAD (*Guo et al., 2020*; *Sun et al., 2020*). Although many studies have identified hundreds of differentially expressed genes (DEGs) and indicated corresponding biological pathways associated with lung cancer, the results were not consistent because of various reasons (*Long et al., 2019*; *Mao et al., 2019*; *Lu et al., 2020*). Considering the limitation of a single microarray analysis, characterized by limited samples, unbalanced datasets and serious systematic error, we herein conducted an integration analysis of genetic data with multiple gene expression profiles and public databases to overcome the shortage and obtain reliable diagnosis markers.

In this work, multiple gene expression profiles retrieved from the Gene Expression Omnibus (GEO) database (https://www.ncbi.nlm.nih.gov/geo/) were used for identification of DEGs and candidate core genes. Protein-protein interaction (PPI) was then constructed following the identification and validation of the hub gene. In addition, survival analysis, meta-analysis, prognostic analysis, and co-expression analysis were performed successively to evaluate the potential of the core gene as a prognostic factor in LUAD. The biological functions of the hub gene were explored through the Gene Set Enrichment Analysis (GSEA) in our study. The workflow of this work is provided in Fig. 1.

## MATERIALS AND METHODS

### Data collection and processing

Eight gene expression profiles were obtained from the GEO online public database, which was provided in Table S1. The adjusted *p*-value < 0.05 and |$\log_2$(foldchange)| ≥ 2 were set as cut-off values for screening DEGs in GSE18842, GSE19188, GSE75037 and

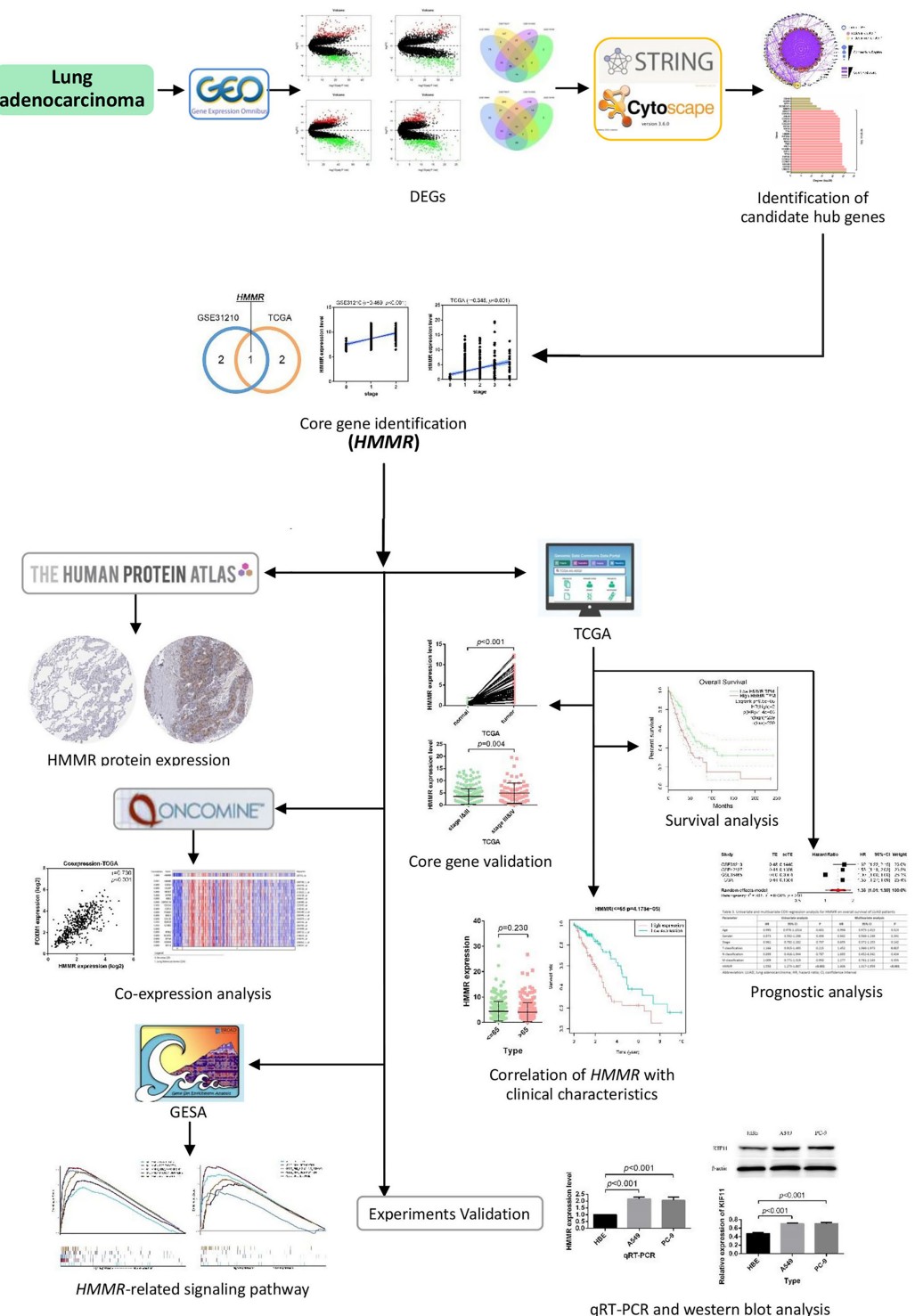

**Figure 1 Flow chart of identification of *HMMR* as a prognostic factor for lung adenocarcinoma (LUAD).**
GSE101929 profiles using R package "limma". Subsequently, the common DEGs shared by the four profiles were used for further study. Additionally, transcriptome RNA-sequencing and proteomics data were extracted from the Cancer Genome Atlas (TCGA) database (https://cancergenome.nih.gov/) andthe Clinical Proteomic Tumor Analysis Consortium (CPTAC) database (https://cptac-data-portal.georgetown.edu/), repetively.

## PPI network construction and enrichment analysis

Construction of PPI network and identification of key module were constructed as previously described in *Li et al. (2021)*. Then, the genes with the highest connectivity degrees in key module were recognized as candidate hub genes, whose mRNA expression levels were further validated *via* GSE19804 and TCGA datasets. The FunRich software (version 3.1.3) was applied to perform enrichment analysis in biological process, cellular component, molecullar function and biological pathway with the cutoff criteria of $p < 0.05$ for the corresponding genes in key module.

## Hub gene identification and validation

Correlations between candidate hub genes and tumor stages were determined by Pearson's correlation analysis in GSE31210 profile and TCGA dataset using R package "corrplot". Then, the common genes shared by the two datasets were identified as hub genes. Furthermore, the protein expression levels of hub geneswere further verified utilizing CPTAC data and immunohistochemical results from Human Protein Atlas (HPA) (https://www.proteinatlas.org). In addition to public datasets, we performed qRT-PCR and western blotting to measure the expression pattern of the core gene in LUAD cells (A549 and PC-9).

## Survival analysis and prognostic value analysis

The association of hub genes with overall survival (OS), progress-free survival (PFS) and disease-free survival (DFS) in LUAD patients were conducted as previously described in *Li et al. (2021)*. Meanwhile, Kaplan-Meier Plotter (www.kmplot.com) and Gene Expression Profiling Interactive Analysis (GEPIA) (http://gepia.cancer-pku.cn/) platforms were also used to perform survival analysis for the corresponding genes. Additionally, the prognostic estimation of hub genes using meta-analysis was performed based on the methods in *Li et al. (2021)*. Finally, Cox regression analysis was conducted following the protocols in *Li, Qi & Li (2020)*.

## Co-expression analysis and gene set enrichment analysis (GSEA)

Co-expression analyses for hub genes were conducted using the Oncomine database (https://www.oncomine.org) in order to identify a significant factor associated with hub genes, which was further verified in both GSE31210 profile and TCGA dataset using R package "corrplot" and GraphPad Prism software (version 7.0). Additionally, GSEA software (version 4.0.3) was utilized to conduct enrichment analysis for DEGs between high and low hub gene expression subgroups to investigate the biological functions of hub gene. FDR < 0.05 and $p < 0.05$ were set as the cut-off criteria.

## Cell culture

HBE, A549 and PC-9 cells were cultured according to protocols in *Li et al. (2021)*.

## Quantitative RT-PCR analysis

Quantitative RT-PCR analysis was conducted following the instructions in *Li et al. (2021)*. The primer sequences are as follows: *HMMR* forward, 5′-ATGATGGCTAAGCAAG AAGGC-3′ and reverse, 5′-TTTCCCTTGAGACTCTTCGAGA-3′; *GAPDH* forward, 5′-GGAGCGAGATCCCTCCAAAAT-3′ and reverse, 5′-GGCTGTTGTCATACTTCTC ATGG-3′.

## Western blot analysis

Westerm blot analysis was subsequently performed according to protocols in *Li et al. (2021)*. The primary antibody against HMMR was purchased from Abcam (Cambridge, UK, diluted 1:1000, ab124729).

## Statistical analysis

The R (version 3.6.0) and Graphpad prism software (version 7.0) were used to perform statistical analysis. Cochran's Q test and Higgin's $I^2$ statistics were applied for the heterogeneity estimation in the meta-analysis in meta-analysis. The statistical differences in the Kaplan-Meier analysis were calculated by log-rank test. Besides, data are presented as the mean ± SD of at least three independent experiments. The $2^{-\Delta\Delta Ct}$ method was utilized to analyze the results of qRT-PCR and Student's t-test was applied to assess the significance of differences between groups. A *p*-value of < 0.05 was considered statistically significant.

# RESULTS

## Identification of DEGs in LUAD

A total of 429, 287, 672, 513 DEGs were identified in GSE18842, GSE19188, GSE75037 and GSE101929, with 194, 105, 223, 174 up-regulated genes and 235, 182, 449, 339 down-regulated genes, respectively (Figs. 2A–2D). Among them, 115 DEGs were shared by the four datasets, including 38 up-regulated genes and 77 down-regulated genes (Figs. 2E, 2F and Table S2).

## Construction of PPI network and module analysis

The shared DEGs were used to construct the PPI network, which included 87 nodes and 399 edges (Fig. 3A). We also identified a key module (cluster 1) with the highest MCODE score (20.92) from the PPI network, which contained 24 genes. Meanwhile, Table S2 provided more details for the 24 genes. Afterwards, these 24 genes were significantly associated with Cell Cycle, Spindle assembly, Cell growth and/or maintenance, Cell communication and Signal transduction (Fig. 3B and Table S3). Centrosome, Microtubule, Spindle microtubule and other cellular components were mainly enriched by the 24 DEGs (Fig. S1A and Table S4). Besides, the corresponding DEGs mainly enriched in Motor activity, Kinase binding, Protein serine/threonine kinase activity and other molecullar

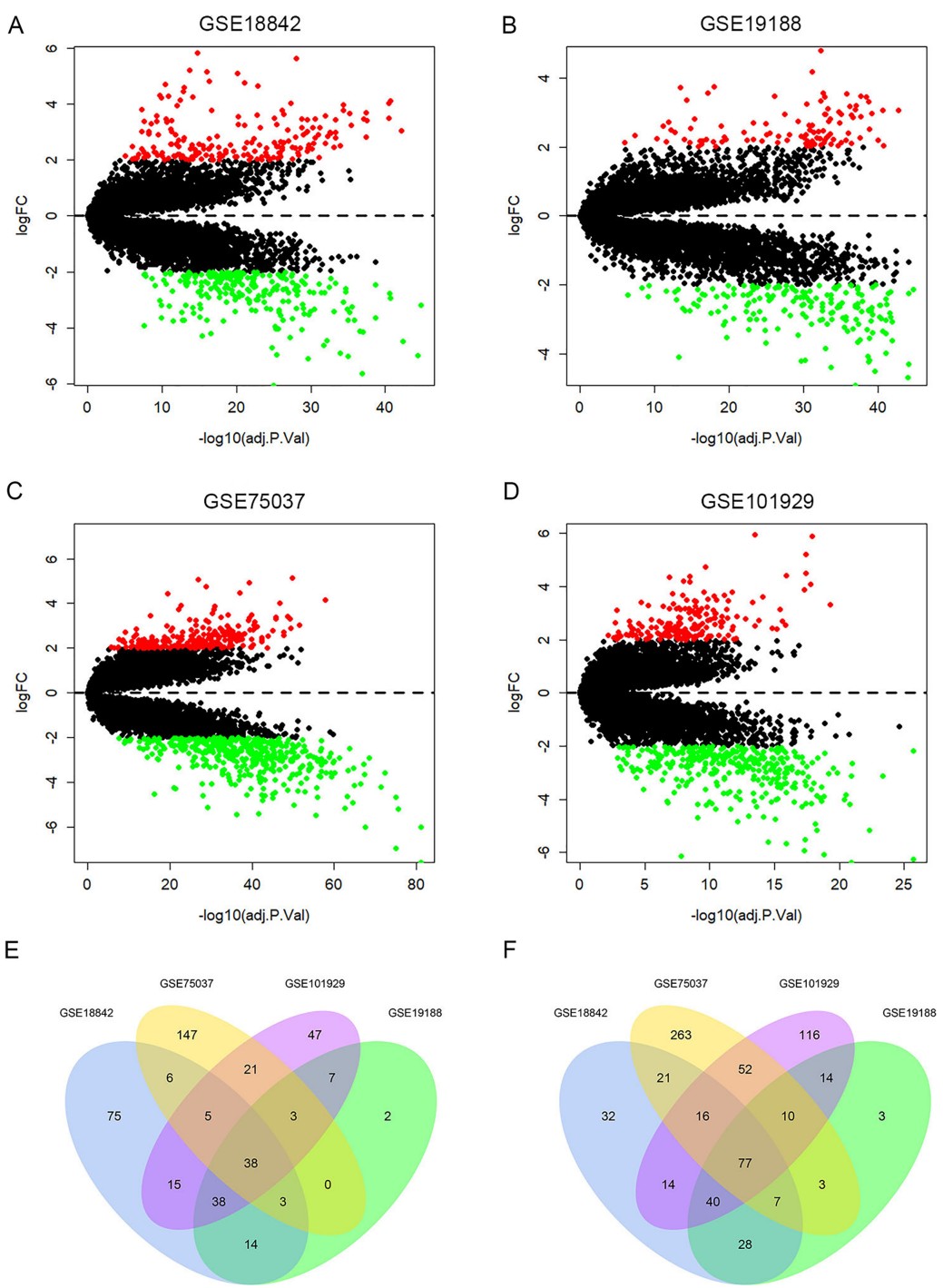

**Figure 2 Identification of differentially expressed genes (DEGs).** Up-regulated (red-colored spots) and down-regulated (green-colored spots) DEGs in LUAD compared with normal lung tissues were identified from four GEO profiles GSE18842 (A), GSE19188 (B), GSE75037 (C) and GSE101929 (D). The 38 up-regulated (E) and 77 down-regulated DEGs (F) were collectively shared by the four GEO expression profiles.                                             

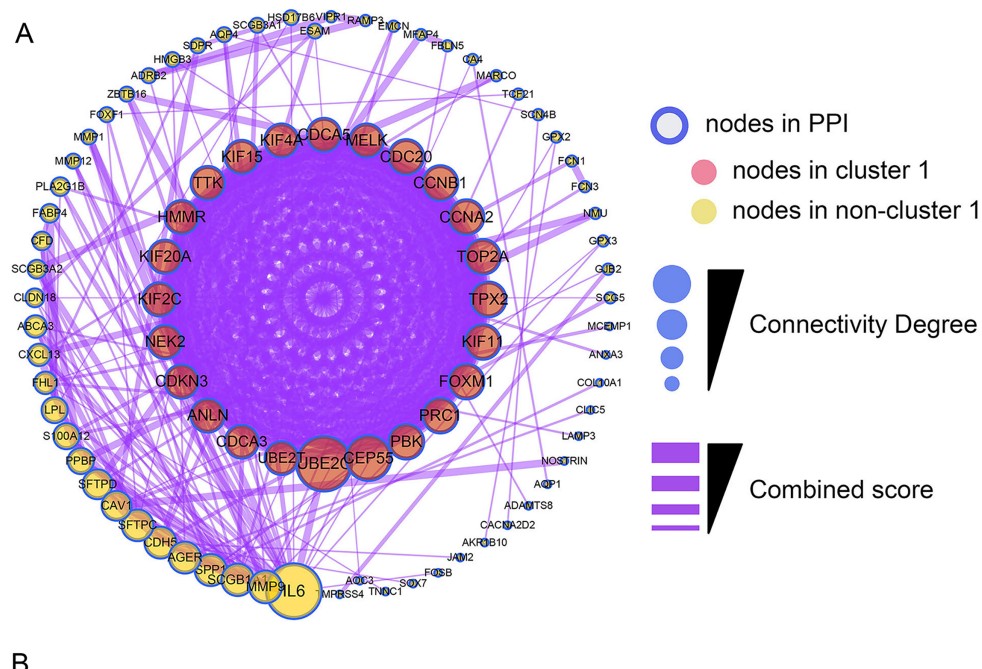

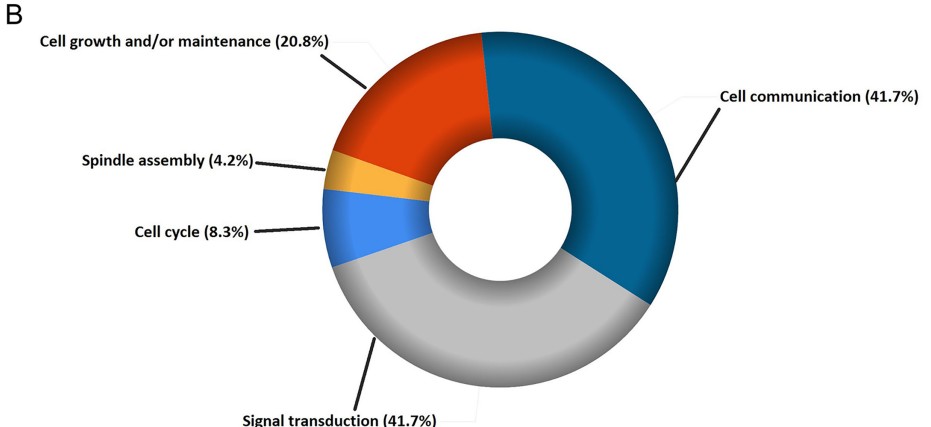

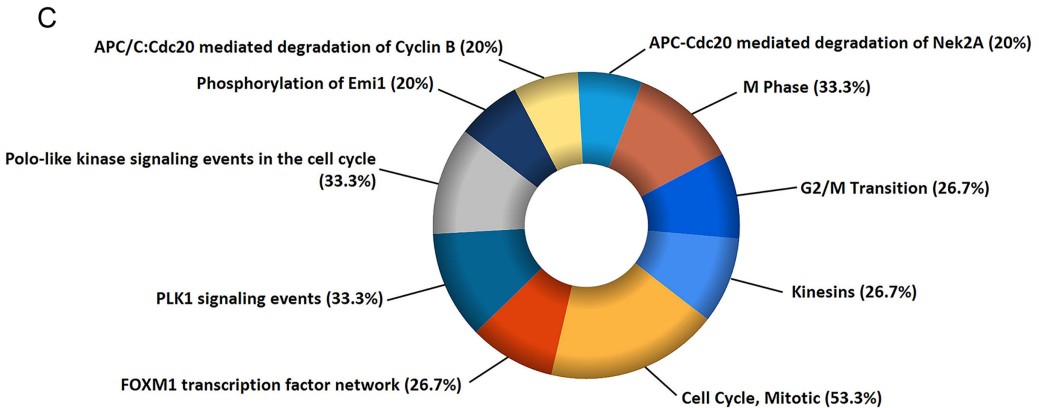

**Figure 3 Construction of protein-protein interaction (PPI) network and module analysis.** (A) The 115 shared DEGs were applied to construct PPI network and perform module analysis. Pink nodes represent the key module (cluster 1), MCODE score = 20.92. Biological process analysis (B) and biological pathway analysis (C) for the corresponding DEGs in the key module.

**Table 1 Correlation of candidate hub genes with tumor stage.**

| GSE31210 profile | | | | TCGA dataset | | | |
|---|---|---|---|---|---|---|---|
| NO. | Gene | R (stage) | P | NO. | Gene | R (stage) | P |
| 1 | HMMR | 0.469 | <0.001 | 1 | CEP55 | 0.235 | <0.001 |
| 2 | KIF20A | 0.449 | <0.001 | 2 | KIF11 | 0.232 | <0.001 |
| 3 | MELK | 0.448 | <0.001 | 3 | HMMR | 0.228 | <0.001 |
| 4 | CCNB1 | 0.447 | <0.001 | 4 | PRC1 | 0.198 | <0.001 |
| 5 | TOP2A | 0.445 | <0.001 | 5 | CCNB1 | 0.194 | <0.001 |
| 6 | KIF2C | 0.443 | <0.001 | 6 | KIF4A | 0.193 | <0.001 |
| 7 | TPX2 | 0.433 | <0.001 | 7 | CDCA5 | 0.185 | <0.001 |
| 8 | CDC20 | 0.432 | <0.001 | 8 | PBK | 0.176 | <0.001 |
| 9 | KIF4A | 0.429 | <0.001 | 9 | MELK | 0.170 | <0.001 |
| 10 | CEP55 | 0.423 | <0.001 | 10 | KIF20A | 0.169 | <0.001 |
| 11 | TTK | 0.417 | <0.001 | 11 | TTK | 0.159 | <0.001 |
| 12 | CDCA5 | 0.417 | <0.001 | 12 | CDKN3 | 0.151 | <0.001 |
| 13 | KIF11 | 0.409 | <0.001 | 13 | TOP2A | 0.141 | 0.002 |
| 14 | CDKN3 | 0.405 | <0.001 | 14 | CCNA2 | 0.137 | 0.003 |
| 15 | CCNA2 | 0.380 | <0.001 | 15 | CDC20 | 0.129 | 0.005 |
| 16 | UBE2C | 0.376 | <0.001 | 16 | UBE2C | 0.106 | 0.021 |
| 17 | PRC1 | 0.372 | <0.001 | 17 | TPX2 | 0.106 | 0.021 |
| 18 | PBK | 0.365 | <0.001 | 18 | KIF2C | 0.098 | 0.033 |
| 19 | KIF15 | 0.364 | <0.001 | 19 | KIF15 | 0.098 | 0.033 |

**Note:**
   R, correlation coefficient; P, *p*-value.

functions (Fig. S1B and Table S5). Meanwhile, Kinesins, Cell Cycle, *FOXM1* transcription factor network were the main enriched biological pathways for these 24 DEGs (Fig. 3C and Table S6).

## Identification of *HMMR* as a hub gene in LUAD

Besides, among the 24 DEGs with high connectivity degrees in the whole PPI network (Figs. 3A and 4A), 19 DEGs were identified with leading intramodular connectivity according to the key module (cluster 1) PPI network (pink nodes constructing the network in Fig. 3A) and meanwhile, they were considered as the candidate hub genesin the work (Fig. 4B). Afterwards, the candidate hub genes were identified in the GSE19804 and GSE31210 profiles (Figs. 4C and 4D). In addition to the difference analysis, the GSE31210 profile and the TCGA dataset were used to evaluate the correlation between the expression of the candidate hub genes and tumor stages (Table 1). The results revealed that *HMMR* was uniquely shared by these two datasets among the top three genes most strongly correlated with tumor stages (Fig. 5A). Though *p*-values were significant, R values in correlation analyses of *HMMR* expression and tumor stage were not satisfied. Therefore, we further performed differential analysis for *HMMR* expression in LUAD patients with different tumor stage aiming to supply the unsatisfied correlation. In the work, *HMMR* was considered as the hub gene. It was highly expressed in LUAD tissues

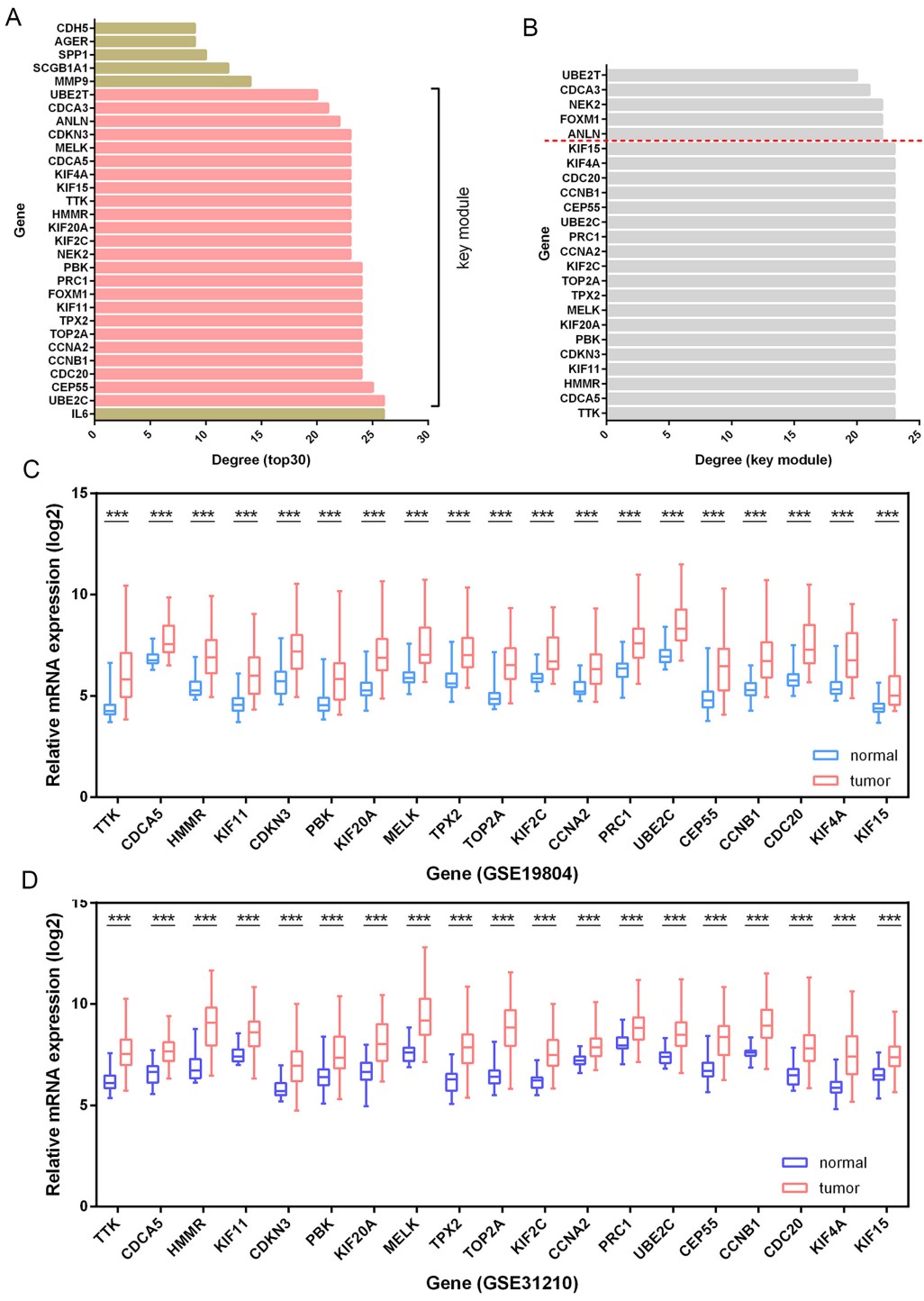

**Figure 4 Identification of candidate hub genes.** (A) The 30 genes with the highest connectivity degrees in the PPI network. The 19 DEGs (candidate hub genes) with the highest connectivity degrees in the key module (B) were further validated in the GSE19804 (C) and GSE31210 (D) profiles. ***, *p*-value<0.001.

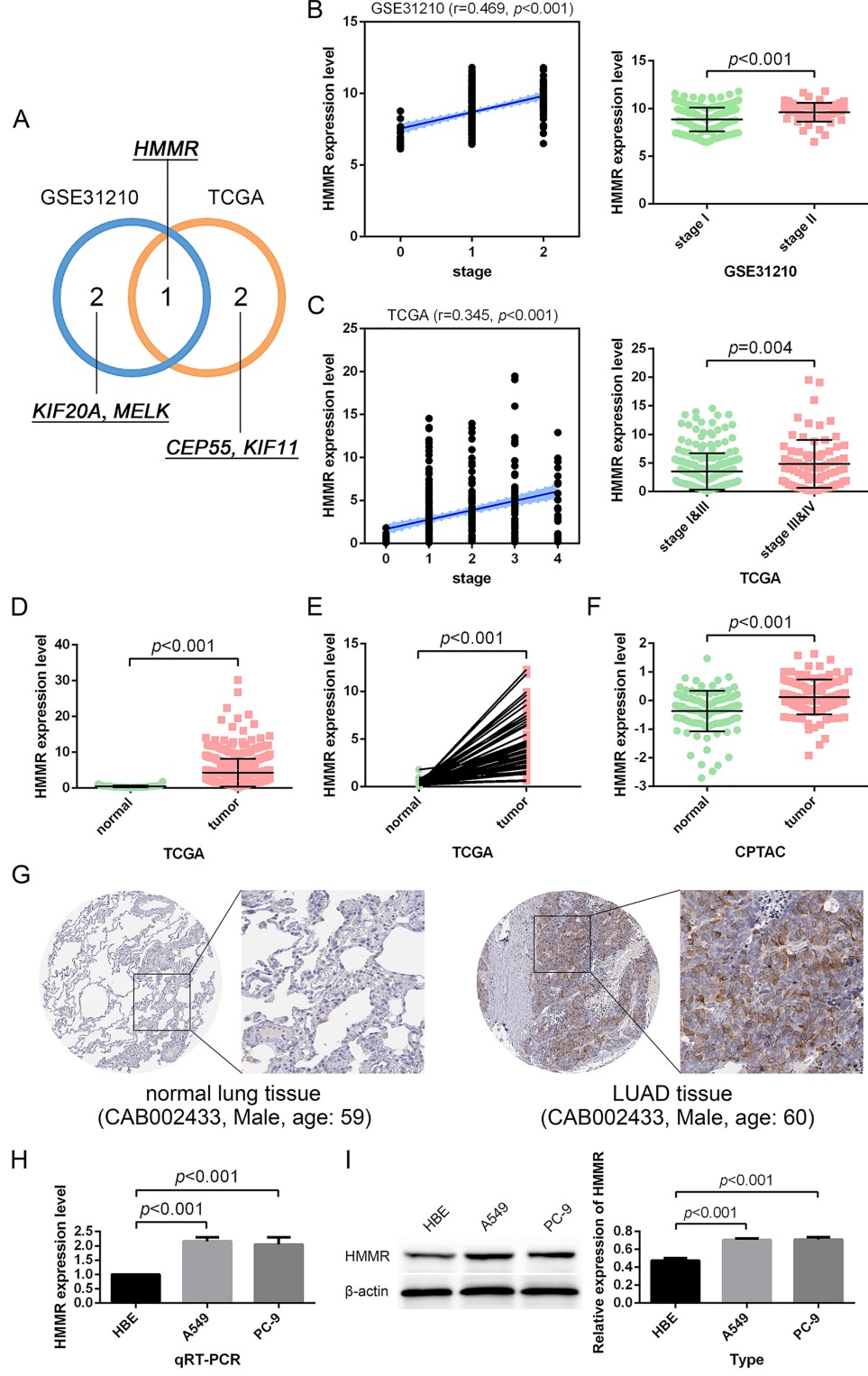

**Figure 5 Identification and validation of the hub gene.** (A) *HMMR* was uniquely shared by the GSE31210 profile and the TCGA dataset among the top three strongly-correlated factors. (B, C) *HMMR* was significantly associated with tumor stage. (D–G) *HMMR* showed remarkably high expression in LUAD tissues compared with normal lung tissues. The expression of *HMMR* was further determined by the qRT-PCR (H) and western blot (I) analyses.

compared with normal lung tissues according to non-paired (Fig. 5D) and paired (Fig. 5E) statiscal analysis in TCGA dataset. Similarly, the protein expression level of HMMR was signifcantly up-regulated in LUAD samples *vs* normal samples based on CPTAC (Fig. 5F) and HPA (Fig. 5G) datasets. Furthermore, the results were further validated using qRT-PCR and western blot analyses (Figs. 5H and 5I).

### *HMMR* served as a prognostic factor in LUAD

It was found that high *HMMR* expression significantly deteriorated overall survival (OS) in TCGA dataset, GSE68465 profile, GSE50081 profile, Kaplan-Meier Plotter and GEPIA platforms (Figs. 6A–6E), as presented in GSE31210 profile (Fig. S2). Besides, *HMMR* was negatively associated with progression-free survival (PFS) in TCGA dataset, GSE68465 profile, Kaplan-Meier Plotter platform (Figs. 6F–6H). The associations of *HMMR* with inferior disease-free survival (DFS) were also observed in Figs. 6I and 6J using GSE50081 profile and GEPIA platform. Because of the unsignificant heterogeneity ($I^2 < 50\%$, $p > 0.05$), we selected solid model to perform the meta-analysis. The results of the meta-analysis implied that *HMMR* served as a prognostic factor for the OS of LUAD patients (Fig. 6K). Moreover, the Cox regression analysis revealed that *HMMR* acted as an independent prognostic factor for the survival of LUAD patients (Table 2).

### Prognostic value of *HMMR* in LUAD

The *HMMR* expression was significantly impacted by gender (female *vs.* male), tumor stage (stage I & II *vs.* stage III&IV), T classification (T1 *vs.* T2-4), N classification (N0 *vs.* N1-3), and M classification (M0 *vs.* M1), whereas not by age (<=65 *vs.* >65) (Fig. 7A). The LUAD patients were classified into high and low *HMMR* expression subgroups according to the median of *HMMR* expression levels. Subsequent survival analysis showed that high *HMMR* expression group exhibited significantly poor OS in >65 age, female, male, stage III&IV, M0 and N1-3 subgroups (Fig. 7B). Interestingly, there was no statistical significance for *HMMR* expression between >65 age subgroup and <=65 age subgroup, while expression level of *HMMR* significantly impacted the OS for LUAD patients with above 65 age. To some extent, there was contradictory for the results might due to limited samples and unbalanced data, which was needed further investigation.

### *HMMR* co-expressed with *FOXM1* in LUAD

According the results of pathway enrichment analysis, we have identified the *FOXM1* transcription factor network and cell cycle were the main biological pathway enriched by the common 24 DEGs (Fig. 3C). Among the pathways, FOXM1 (forkhead box protein M1) was an essential proliferation-associated transcription factor. It was widely expressed during cell cycle and involved in cellular growth, self-renewal, and tumorigenesis (*Liao et al., 2018*). Additionally, through Oncomine co-expression analysis, we found that the expression of *HMMR* was positively associated with that of *FOXM1* (r = 0.890) (Fig. 8A), which was consistent with our expectation. The mRNA expression of *FOXM1* exhibited significant elevation in LUAD tissues relative to the matched normal lung tissues in both the GSE31210 profile and the TCGA data. In line with Oncomine database, the GSE31210 profile and the

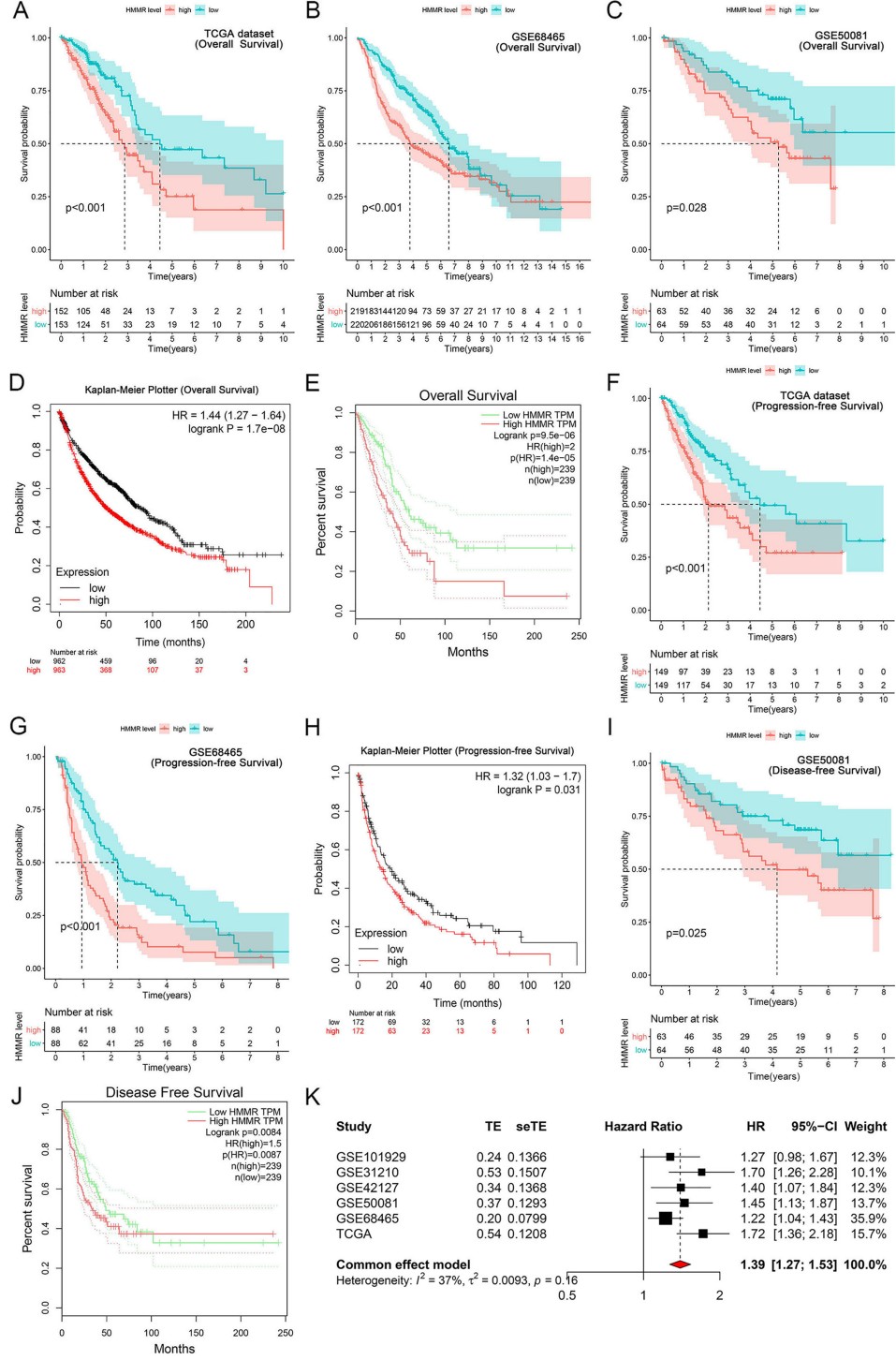

**Figure 6 Correlation of *HMMR* expression with the survival of LUAD patients.** *HMMR* was negatively associated with overall survival (OS) (A–C), progression-free survival (PFS) (D, E) and disease-free survival (DFS) (F) in LUAD patients. Kaplan-Meier Plotter (G, H) and GEPIA (I, J) datasets showed that high *HMMR* expression significantly deteriorated OS, PFS, and DFS in LUAD patients. (K) Meta-analysis suggested that *HMMR* could serve as a prognostic factor for the OS of LUAD patients.

**Table 2 Cox analysis of *HMMR* on OS and PFS in LUAD patients.**

| Parameter | Univariate Cox analysis | | | Multivariate Cox analysis | | |
|---|---|---|---|---|---|---|
| | HR | 95% CI | *P* | HR | 95% CI | *P* |
| **TCGA (OS)** | | | | | | |
| *HMMR* | 1.550 | [1.273–1.887] | **<0.001** | 1.606 | [1.317–1.959] | **<0.001** |
| Age | 0.995 | [0.976–1.014] | 0.601 | 0.994 | [0.975–1.013] | 0.523 |
| Gender | 0.873 | [0.592–1.288] | 0.494 | 0.842 | [0.568–1.248] | 0.391 |
| Stage | 0.961 | [0.782–1.182] | 0.707 | 0.655 | [0.372–1.153] | 0.142 |
| T classification | 1.166 | [0.915–1.485] | 0.215 | 1.452 | [1.068–1.973] | **0.017** |
| M classification | 0.899 | [0.416–1.944] | 0.787 | 1.693 | [0.452–6.342] | 0.434 |
| N classification | 1.009 | [0.771–1.319] | 0.950 | 1.277 | [0.761–2.143] | 0.355 |
| **TCGA (PFS)** | | | | | | |
| *HMMR* | 1.627 | [1.315–2.014] | **<0.001** | 1.693 | [1.368–2.094] | **<0.001** |
| Age | 0.992 | [0.973–1.012] | 0.429 | 0.993 | [0.973–1.013] | 0.491 |
| Gender | 0.825 | [0.557–1.221] | 0.336 | 0.800 | [0.536–1.194] | 0.274 |
| Stage | 0.870 | [0.703–1.075] | 0.197 | 0.641 | [0.359–1.146] | 0.134 |
| T classification | 1.075 | [0.832–1.389] | 0.581 | 1.371 | [0.978–1.921] | 0.067 |
| M classification | 0.610 | [0.247–1.508] | 0.285 | 1.556 | [0.347–6.972] | 0.564 |
| N classification | 0.910 | [0.700–1.185] | 0.485 | 1.163 | [0.696–1.945] | 0.565 |
| **GSE68465 (OS)** | | | | | | |
| *HMMR* | 1.001 | [1.000–1.002] | **0.002** | 1.001 | [1.000–1.002] | **0.045** |
| Gender | 1.427 | [1.101–1.849] | **0.007** | 1.265 | [0.971–1.648] | 0.081 |
| Age | 1.027 | [1.013–1.040] | **<0.001** | 1.029 | [1.015–1.043] | **<0.001** |
| Grade | 1.135 | [0.934–1.379] | 0.204 | 0.960 | [0.769–1.199] | 0.719 |
| N classification | 2.012 | [1.712–2.365] | **<0.001** | 1.982 | [1.683–2.335] | **<0.001** |
| T classification | 1.665 | [1.387–1.998] | **<0.001** | 1.423 | [1.174–1.725] | **<0.001** |
| **GSE68465 (PFS)** | | | | | | |
| *HMMR* | 1.002 | [1.001–1.003] | **0.001** | 1.001 | [1.000–1.003] | **0.012** |
| Gender | 1.224 | [0.872–1.718] | 0.243 | 1.230 | [0.871–1.736] | 0.240 |
| Age | 1.010 | [0.991–1.030] | 0.305 | 1.017 | [0.998–1.037] | 0.085 |
| grade | 1.492 | [1.125–1.979] | **0.006** | 1.208 | [0.892–1.634] | 0.222 |
| N classification | 1.559 | [1.257–1.934] | **<0.001** | 1.570 | [1.251–1.971] | **<0.001** |
| T classification | 1.498 | [1.175–1.909] | **0.001** | 1.253 | [0.961–1.635] | 0.096 |

**Note:**
LUAD, lung adenocarcinoma; HR, hazard ratio; CI, confidence interval; P, *p*-value; OS, overall survival; PFS, progression-free survival.
The bold entries indicated the statistical significance (*p*-value<0.05).

TCGA data also suggested that the expressions of *HMMR* and *FOXM1*were positively correlated *via* Pearson's correlation analysis, even with similar correlation strengths, with r = 0.723, *p* < 0.001 and r = 0.730, *p* < 0.001, respectively (Figs. 8B, 8C).

## *HMMR* was involved in cell cycle in LUAD

GSEA was performed to illuminate the potential biological functions of *HMMR* in LUAD progression using DEGs between high and low *HMMR* subgroups according to the median

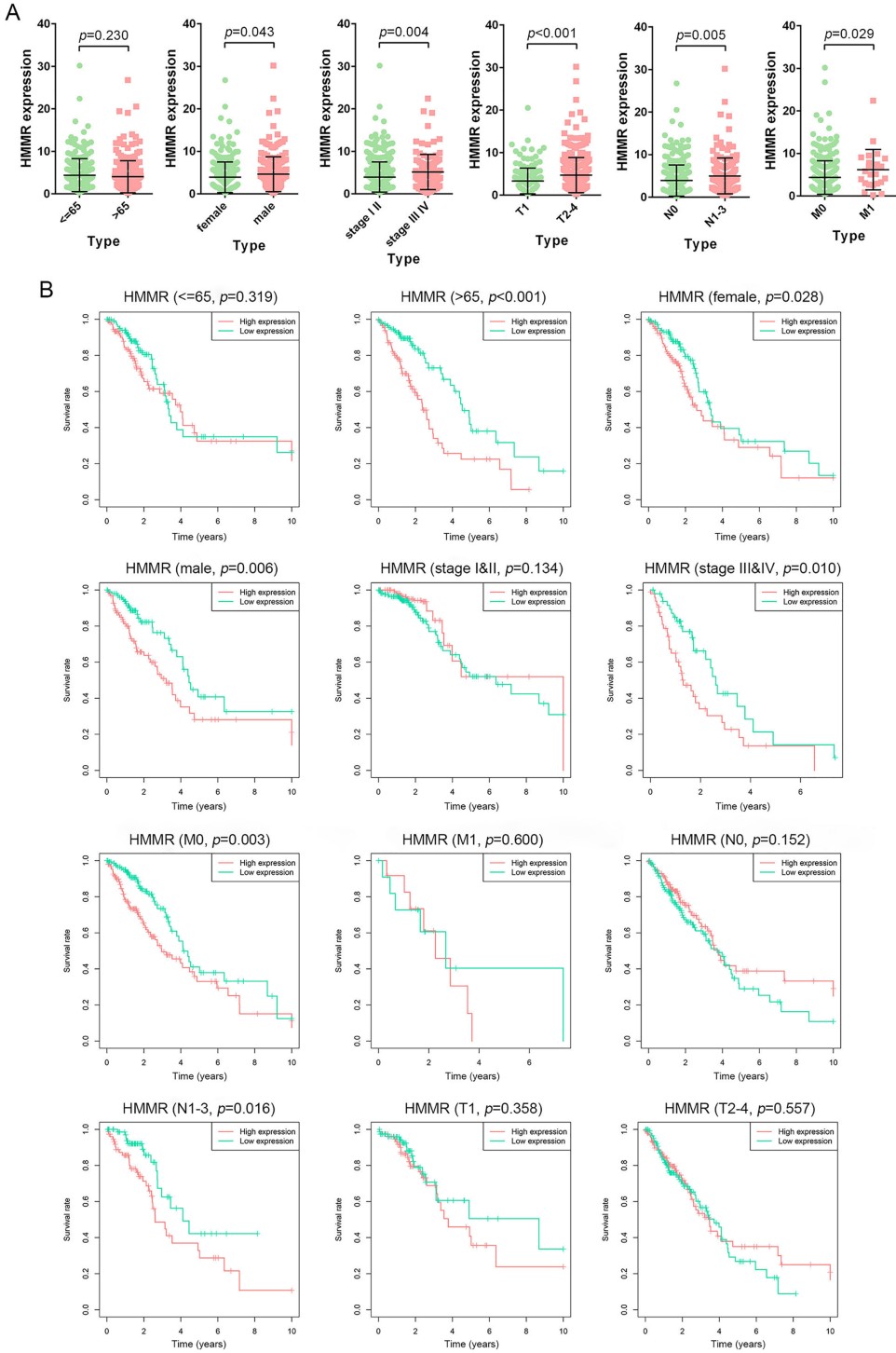

**Figure 7 Prognostic significance of *HMMR* in LUAD.** (A) Difference analysis for the *HMMR* expression in LUAD patients with different demographic and clinical characteristics, including age, gender, tumor stage, primary tumor, lymph node metastasis status, and distant metastasis status. (B) Kaplan-Meier curves for the OS in LUAD patients with specific clinical characteristics. LUAD patients were divided into high and low *HMMR* expression subgroups based on the median of *HMMR* expression levels.

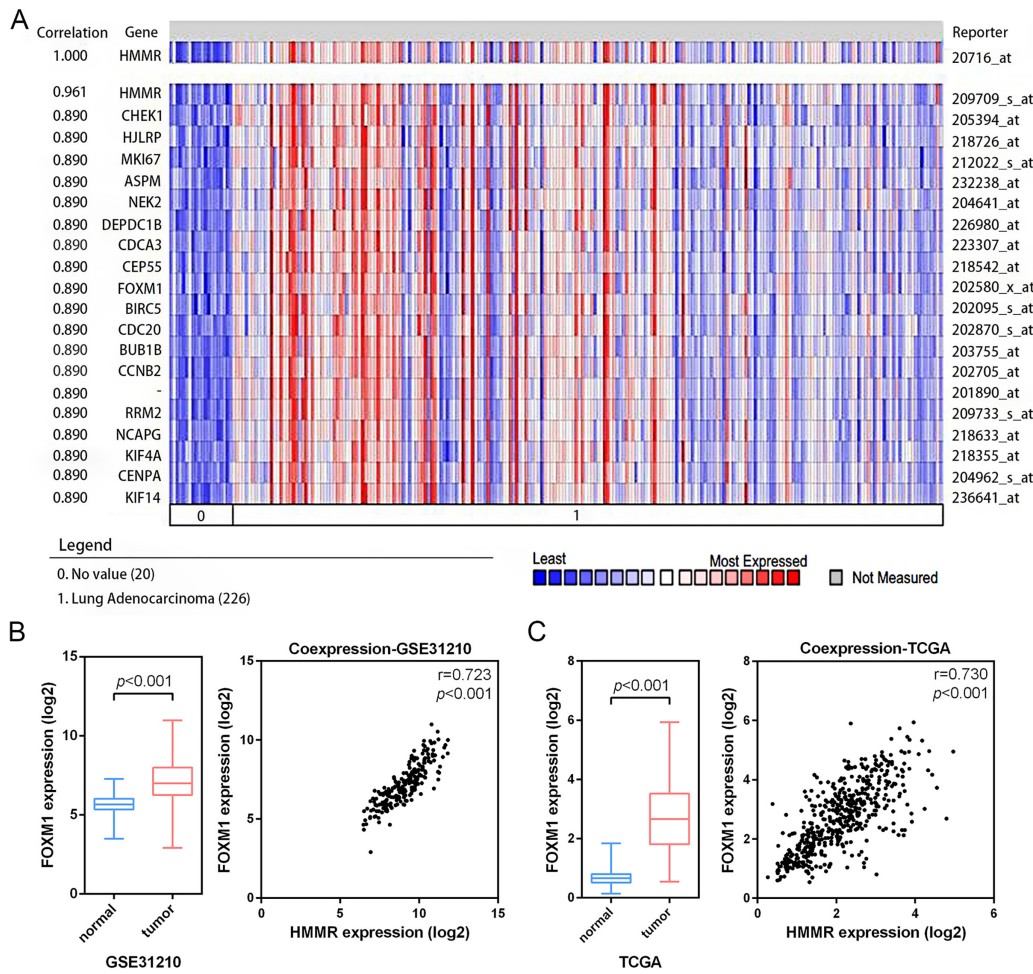

**Figure 8 Co-expression of *HMMR* and *FOXM1*.** A strong co-expression relationship between *HMMR* and *FOXM1* within LUAD was identified in the Oncomine database (A) and validated in the GSE31210 profile (B) and the TCGA dataset (C).               

*HMMR* expression level. As shown in Fig. 9A and Table S7, the functions of these genes were significantly associated with cell cycle, G2/M phase transition, cell cycle process, chromosome segregation, and nuclear division pathways, as indicated by biological process enrichment analysis. DNA repair, E2F targets and G2/M checkpoint were the major enriched HALLMARK terms (Fig. 9B and Table S8). Cell cycle, DNA replication, p53 signaling pathway, and other biological pathways were the major enriched HALLMARK terms (Fig. 9C and Table S9). Taken together, those results indicated that the implication of *HMMR* in LUAD progression might be mediated by cell cycle regulation.

## DISCUSSION

Currently, LUAD is considered to be the most common subtype of lung cancer in clinical practice, with a 5-year overall survival rate varying from 4% to 17% (*Hirsch et al., 2017*). The exact pathogenesis of LUAD remains elusive and numerous diverse and complicated processes have been reported to play a role. For instance, the presence of aberrant gene expression, autophagy activation, unexpected tumor microenvironment,

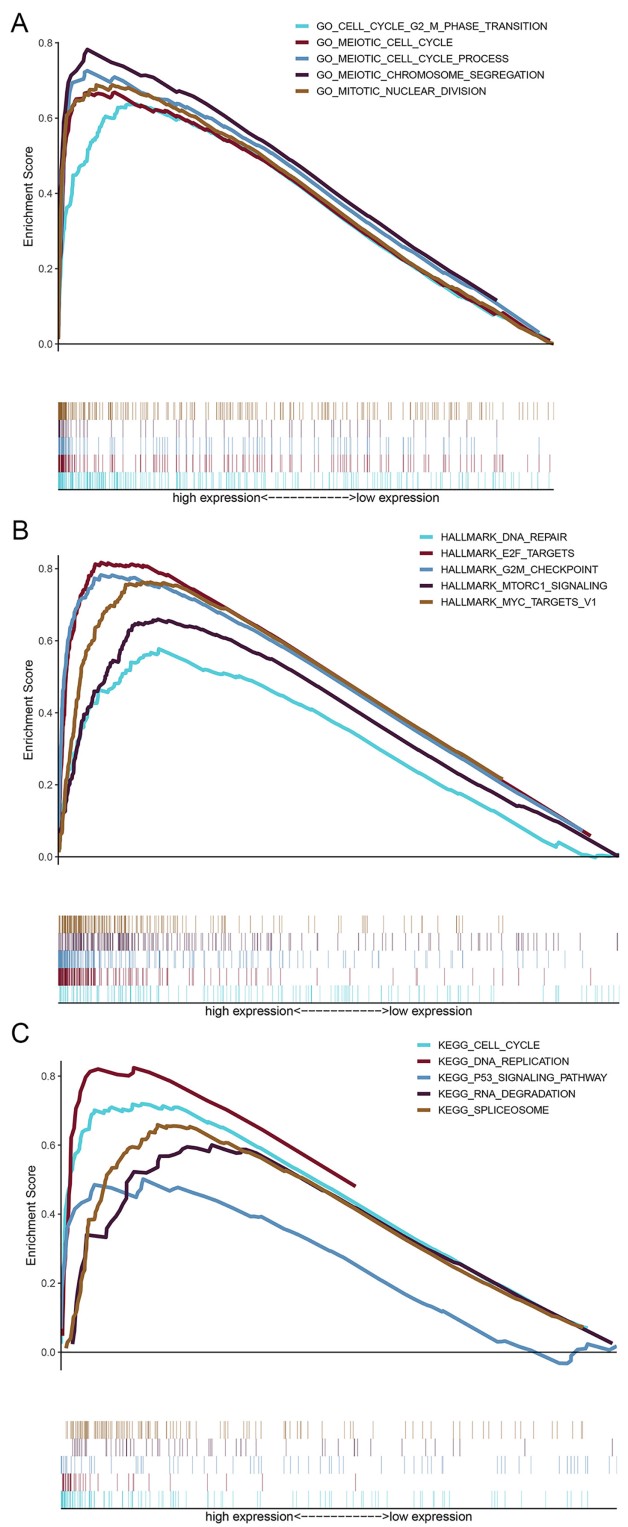

**Figure 9 Gene set enrichment analysis (GSEA) for upregulated genes in the high *HMMR* expression group.** (A) Results of biological process enrichment analysis. (B) Results of HALLMARK enrichment analysis. (C) Results of KEGG enrichment analysis.
immune cell infiltration, abnormal cell cycle, DNA methylation, epigenetic interactions, and other molecular and cellular events, have been identified to be associated with the development of LUAD (*Fan et al., 2019*; *Chen et al., 2019b*; *Wang et al., 2019a*; *Lazarus et al., 2018*). It has also been reported that several signaling pathways, such as MAPK pathway, AKT-mTOR signaling and Wnt/β-catenin signaling pathway, are frequently activated in LUAD (*Hou et al., 2019*; *Liu et al., 2020*; *Luo et al., 2019a*). In spite of the great progress in illustrating the pathogenesis of LUAD and treating LUAD, effective prognostic biomarkers and treatment targets are stilling lacking. Hence, there is an urgent need to find a precise and effective prognostic factor and a therapeutic target in order to improve LUAD-associated survival rate.

Bioinformatics is a data-driven subtype of science and currently frequently applied in aspects like analyzing genetic data, studying tumor progression, screening core genes, and identifying the medical targets through the usage of bioinformatic analytical tools including software plug-ins and packages, as well as database platforms (*Luo et al., 2019b*; *Li et al., 2019*). Previously, an integrated regulatory network, containing DEGs, transcription factors, and miRNAs, has been constructed using GSE37764 dataset, which attempted to find out the important factors in the pathogenesis of lung adenocarcinoma and lay the foundation for the further investigation (*Du & Zhang, 2015*). Meanwhile, an eight-gene prognostic signature (*DLGAP5*, *KIF11*, *RAD51AP1*, *CCNB1*, *AURKA*, *CDC6*, *OIP5* and *NCAPG*) also has been identified for the survival of LUAD using GSE19188 and GSE33532 profiles (*Li et al., 2018*). Besides, *Zhou et al. (2019)* aimed to screen out the driver genes in smoking-associating lung adenocarcinoma *via* bioinformatics, and then they suggested that the seven genes (*CYP17A1*, *PKHD1L1*, *RPE65*, *NTSR1*, *FETUB*, *IGFBP1* and *G6PC*) could be promising prognostic factors for lung adenocarcinoma according to their negative correlation with the patient survival.

In this study, we aimed to find a core gene that has both prognostic value and the potential to become a treatment target for LUAD *via* integrated analysis. Firstly, shared DEGs by the four GEO profiles were obtained and used for PPI network construction followed by key module identification in order to identify candidate hub genes. Secondly, biological process and pathway enrichment analyses for the identified genes were carried out by the FunRich software. The enrichment results suggested that the potential functions of those candidate genes were specifically associated with cell cycle, spindle assembly, *FOXM1* transcription factor network, M phase, and other pathways. It is well known that cell cycle and M phase transition play essential roles in manipulating the proliferation of cells and the occurrence of tumors. For example, repressing p21 expression, the check point of cell cycle, could promote tumor proliferative capacity and accelerate cancer procession (*Zhang et al., 2019a*). Likewise, many molecules, like Rac3, SMAD3, CDCA7 and DMBX1, which regulate cell growth *via* cell cycle, have similar functions (*Li et al., 2017*; *Wang et al., 2016a*, *Wang et al., 2019b*; *Luo et al., 2019c*). Accumulating studies have reported the involvement of *FOXM1* in the progression of tumors including lung cancer, gastric cancer, breast cancer, and epithelial ovarian cancer (*Hsieh et al., 2019*; *Bai et al., 2019b*; *Ring et al., 2018*; *Wang et al., 2016b*). Specifically, in lung cancer, *FOXM1* has been found to be co-expressed with *CENE* and could regulate the

expression of *MMP2*, contributing to LUAD growth and metastasis (*Shan et al., 2019*; *Hsieh et al., 2019*). Besides, supporting our findings in the present study, *FOXM1* transcription factor network was identified as a major predictor of poor outcomes in pan-cancer in the report of *Gentles et al. (2015)*. Finally, among these candidate genes, *HMMR* was identified as the hub gene in this study due to its significant role in predicting survival and its association with tumor stage, which was further validated at the protein level in the HPA database and at the mRNA expression level on the CCLE platform. The potential functions of the hub gene were explored by GSEA, and the results indicated that mainly cell cycle and its relevant pathways were activated pathways in LUAD highly expressing *HMMR*.

The above results confirmed the prognostic value of *HMMR* expression in LUAD. HMMR, a sulfonated glycosaminoglycan, is a receptor for hyaluronic acid (HA), which accumulated during pulmonary inflammation. It has been found that high expression of *HMMR* is associated with multiple human malignancies, such as gastric cancer, breast cancer, prostate cancer, ovarian cancer, bladder cancer, with characteristics of promoting cancer progression and indicating poor prognosis in patients (*Huang et al., 2017*; *Yeh et al., 2018*; *Rizzardi et al., 2014*). For example, a zebrafish xenograft assay verified that highly expressed *HMMR*, under the control of both TGFβ signaling and Hippo pathway, contributed to sarcoma genesis and metastasis (*Ye et al., 2020*). In gastric cancer patients, *HMMR* over-production was remarkably associated with tumor relapse and poor prognosis, and resulted in resistance to the chemotherapy *via* promoting epithelial-mesenchymal transition and modulating cancer stem cell properties (*Zhang et al., 2019b*). Additionally, HMMR also interacted with CD44, another HA receptor characterized by forming complexes with ERK1/2, to exert its functions in breast cancer (*Telmer et al., 2011*). Besides, HMMR was identified as a promising diagnostic biomarker and an independent prognostic factor for hepatocellular carcinoma (HCC) *via* bioinformatics, the expression of which was positively correlated with HCC tumor grade and stage in HCC patients (*Lu et al., 2020*).

We found that high expression of *HMMR* was significantly correlated with poorer OS and survival rate in subgroups with clinical stage III/IV, lymph node metastasis classification 1/2/3, male, female and patients with an age of above 65. Multivariate COX regression analysis further suggested that the *HMMR* expression level could serve as an independent prognostic indicator in LUAD. These findings highlighted the prognostic value of *HMMR* expression in LUAD. We further conducted correlation analysis based on the Oncomine database and the results indicated the expression of *HMMR* was positively associated with that of *FOXM1*, which encoded transcription factor of fork head family and intriguingly was also found to negatively impact prognosis in many solid tumors (*Liang et al., 2019*; *Liu et al., 2018*). Additionally, cell cycle, DNA replication, p53 signaling pathway, RNA degradation, spliceosome, and ubiquitin mediated proteolysis were significantly enriched pathways in LUAD highly expressing *HMMR*. It was well known that these six signaling pathways were typically involved in the occurrence and development of cancers (*Aubrey et al., 2018*; *Fish et al., 2019*; *Dvinge et al., 2019*; *Senft, Qi & Ronai, 2018*). For example, the core protein and RNA components of the spliceosome

were essential for splicing decision and able to manipulate the splice sites during pre-mRNA processing, thereby playing importantly regulatory roles in tumorigenesis and metastasis of malignant cells (*Hsu et al., 2015*). However, there was no inhibitor of HMMR currently undergoing preclinical or clinical testing anywhere in the world.

In the work, we identified *HMMR* as the core gene *via* integrated bioinformatics analysis and provided robust evidence for the potential prognostic and therapeutic role of *HMMR* in LUAD, though the prognostic value was not significant in patients with clinical stage I/II and an age of below 65. It might also be a limitation, as only RNA-based bioinformatics analysis was performed in this study and no functional experiment was conducted. Therefore, further studies are needed to improve the reliability of the results.

## CONCLUSION

In summary, *HMMR* might serve as an independent prognostic factor for the OS in LUAD patients. This work would facilitate the development of novel prognostic biomarkers and therapy targets for LUAD, though further experiments are needed to verify these findings.

## ACKNOWLEDGEMENTS

We would like to thank TopEdit for its linguistic assistance during the preparation of this manuscript. Besides, we are extremely grateful for reviewers' input in helping this manuscript.

### Funding

The authors received no funding for this work.

### Competing Interests

The authors declare that they have no competing interests.

### Author Contributions

- Zhaodong Li conceived and designed the experiments, performed the experiments, analyzed the data, prepared figures and/or tables, authored or reviewed drafts of the paper, and approved the final draft.
- Hongtian Fei conceived and designed the experiments, performed the experiments, analyzed the data, prepared figures and/or tables, and approved the final draft.
- Siyu Lei performed the experiments, analyzed the data, prepared figures and/or tables, and approved the final draft.
- Fengtong Hao performed the experiments, analyzed the data, prepared figures and/or tables, and approved the final draft.
- Lijie Yang performed the experiments, analyzed the data, prepared figures and/or tables, and approved the final draft.
- Wanze Li performed the experiments, analyzed the data, prepared figures and/or tables, and approved the final draft.

- Laney Zhang analyzed the data, prepared figures and/or tables, authored or reviewed drafts of the paper, and approved the final draft.
- Rui Fei conceived and designed the experiments, performed the experiments, analyzed the data, authored or reviewed drafts of the paper, and approved the final draft.

## Data Availability

- TCGA-LUAD: https://portal.gdc.cancer.gov/repository?facetTab=cases&filters=%7B%22op%22%3A%22and%22%2C%22content%22%3A%5B%7B%22op%22%3A%22in%22%2C%22content%22%3A%7B%22field%22%3A%22cases.disease_type%22%2C%22value%22%3A%5B%22adenomas%20and%20adenocarcinomas%22%5D%7D%7D%2C%7B%22op%22%3A%22in%22%2C%22content%22%3A%7B%22field%22%3A%22cases.primary_site%22%2C%22value%22%3A%5B%22bronchus%20and%20lung%22%5D%7D%7D%2C%7B%22op%22%3A%22in%22%2C%22content%22%3A%7B%22field%22%3A%22cases.project.program.name%22%2C%22value%22%3A%5B%22TCGA%22%5D%7D%7D%2C%7B%22op%22%3A%22in%22%2C%22content%22%3A%7B%22field%22%3A%22cases.project.project_id%22%2C%22value%22%3A%5B%22TCGA-LUAD%22%5D%7D%7D%5D%7D

- NCBI GEO, GSE18842, GSE19188, GSE75037, GSE101929, GSE19084, GSE31210, GSE50081 and GSE68465;

- HPA:

- https://www.proteinatlas.org/ENSG00000072571-HMMR/pathology/lung+cancer#img

- https://www.proteinatlas.org/ENSG00000072571-HMMR/tissue/lung#img

- Kaplan-Meier Plotter: https://kmplot.com/analysis/index.php?p=service&cancer=lung

- GEPIA, HMMR: http://gepia.cancer-pku.cn/detail.php?gene=HMMR&clicktag=survival.

## Supplemental Information

Supplemental information for this article can be found online at http://dx.doi.org/10.7717/peerj.12624#supplemental-information.

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
