# Peer review of "Identification of HMMR as a prognostic biomarker for patients with lung adenocarcinoma via integrated bioinformatics analysis"

_PeerJ, doi:10.7717/peerj.12624_

## Round 0.1 · original submission · Major Revisions

The Reviewers made several points regarding the reporting and the design of the study itself and those need to be addressed to clarify the story. Especially remarks on choosing datasets and citing/commenting on other research communicated to date.

Also, there are several papers on DEGs in LUAD which are not discussed here, for instance, to list the three first:

1) Identification of an eight-gene prognostic signature for lung adenocarcinoma (https://doi.org/10.2147/CMAR.S173941) which used GSE19188 and GSE33532 datasets

2) Integrated analysis of DNA methylation and microRNA regulation of the lung adenocarcinoma transcriptome (https://doi.org/10.3892/or.2015.4023) describing GSE37764 dataset

3) Bioinformatics and functional analyses of key genes in smoking-associated lung adenocarcinoma (https://doi.org/10.3892/ol.2019.10733) based on Cancer Genome Atlas database

Although those might have slightly other assumptions and aims than your project, they provide valuable data on differentially expressed genes in LUAD and comparing your results with theirs would be only beneficial. Therefore, please consider re-doing the literature search and provide extensive discussion.

Reviewer 1 ·

Basic reporting

In the manuscript titled "Identification of HMMR as a prognostic biomarker..." Li et. al have shown methodically and logically that HMMR might serve as a potential independent prognostic factor for LUAD patients. The paper is well written with proper literature references and background/context. The writing is easy to follow and their conclusions are logical and accurately based on their analyses and observations. I have only some minor suggestions which I think will enhance the quality of the manuscript.

Experimental design

Their questions were well defined and experiments were designed accordingly. Methods have been decribed in sufficient details in my opinion.

Validity of the findings

For all experiments and analyses the conclusions are logical and well stated at the end of the experiment/analysis. All data have been provided and they appear robust and statistically sound.

Additional comments

I have the following minor comments for the authors:
1. For figure 4B, could you please explain how you identified 19 DEGs from 24 as candidate hub genes (line 161-163). Its not clearly stated there.
2. In figure 5A, please mention the names of the 4 proteins in TCGA and GSE31210 that do not overlap with each other. It will be interesting to show (encouraged) the corresponding data (similar to 5B-G) for one or more of these 4 proteins as controls.
3. Please state clearly the difference between figure 5D and E in the text and the legend.
4. Line 171 there seems to be a mislabeling. Fig 5E and F should be H and I.
5. In figure 6F, why DFS has been provided instead of PFS like in the other datasets. If DFS is shown for one, it should be shown for ther three datasets too. Please mae the figure panels consistent with each other so that its easier for readers to follow the figures.
6. The whole section of "HMMR co-expressed with FOXM1" starting from line 191 seems out of context and sudden as no mention of FOXM1 has been made prior to this in the results section. So a few lines should be added in the beginning of the section stating why FOXM1 was selected for testing its correlation with HMMR.
7. Is there any inhibitor for HMMR currently undergoing preclinical/clinical testing anywhere in the world? Would be nice to comment a few lines on this in the discussion section.

·

Basic reporting

Zhaodong Li et al has done an integrated bioinformatics analysis to identify HMMR as a prognostic biomarker for patients with lung adenocarcinoma. Considering the severity of LUAD, this is a very interesting study to identify biomarkers using unbiased LUAD datasets.

Comments to improve:
1)Though the manuscript is written with commendable language, references to previous study in liver/hepatocellular carcinoma with HMMR using similar bioinformatics analysis, like the one published by Donglan Lu in Oncol Lett. 2020 Sep; 20(3): 2645–2654 is not included in introduction or discussion.

2)1) The authors at line 60-63 claim that “many studies have identified hundreds of differentially expressed genes (DEGs) and indicated corresponding biological pathways associated with lung cancer, the results were not consistent because of various reasons”. Reports must be cited to for the above argument. Also, a brief information about the available biomarkers for LUAD and its associated limitations in LUAD would strengthen the case for this study in line 60-63.

Experimental design

The authors have used several reliable and established tools for their analysis. However, there are lot of technical issues with many of the analysis, which must be addressed to strengthen the observations.

Comments to improve
1) Areas like methodology, figure legends etc. are very poorly described, which must be elaborated for easy understanding of the readers. For instance, meta-analysis done in Figure 6K has very minimal information in methodology which makes the readers of different field very difficult to understand. All the experimental design has to be well described in material and methods section.

Validity of the findings

Though similar study is done and HMMR was identified as a good canditate gene in hepatocellular carcinoma using same bioinformatics approach, it is still a meaningful replication and links HMMR with LUAD. However there are many major concerns which should be addressed.

Major Comments to improve:

2) List of genes (115 DEGs) should be included in supplementary as it forms an important piece of data for all the analysis.
3) In Figure 3B and 3C, the criteria used for biological process analysis and pathway analysis must be described in materials section. Also, Figure 3B and C describes the percentage of enrichment. Are these percentages significant? It is more reliable to plot the chart with respect to significant P-values.
4) Additionally, GO analysis (Molecular function and cellular component) will strengthen the authors analysis Figure 3B and C. Also, it’s surprising to see the cancer associated pathways being not enriched in LUNG cancer datasets analyzed in Figure 3C. A more comprehensive and stringent analysis is needed for Figure 3B and 3C.
5) The authors have mentioned about 24 genes in line 154-155. A more detailed interpretation about this key module is required. It is important to elaborate on whether these genes belong to upregulated and/or downregulated or combination of both in results section.
6) The authors selected HMMR based on its shared presence among top three genes strongly correlated with tumor stage. The other hub genes were just omitted, although a strong correlation with tumor stage with very significant p-values were shown in Table 1. Though HMMR looks very interesting gene in LUAD (Figure 5A_C), unfortunately HMMR was selected just by comparing GSE31210 and TCGA. The authors should also include other datasets used in this study (GSE18842, GSE19188, GSE75037 and GSE101929) before concluding HMMR as a strong candidate gene in LUAD.
7) HMMR over expression is very evident in Figure 5B-F between normal and tumor tissue samples. However, when tumor stages were compared with respect to HMMR expression levels, a great correlation was not obtained based on R values, though p-values are significant. The authors may need to discuss this part more elaborately in results section in line 170-171.
8) Figure 5 H and I have no mention in the text. Also, KIF11 qRT-PCR was included, which has not been discussed elsewhere in the manuscript.
9) As HMMR was selected by comparing GSE31210 and TCGA, survival analysis for GSE31210 dataset, if available, is mandated to strengthen the current observations in Figure 5.
10) The authors have calculated P-values at 0.50 survival probability. The authors need to clarify and/or reanalyze why P-values were not calculated for overall survival probability in Figure 6A-F, unlike in Figure 6 G-H and Figure 7B.
11) HMMR and FOXM1 were positively correlated in LUAD. Briefly reporting the known function of FOXM1 in LUAD in line 198-199 would enhance the significance of FOXM1 correlation with HMMR in the current study. Also, FOXM1 was lightly used in the study, an elaborate description of FOXM1 in LUAD is required in results section in line191 -199.
12) Line 201-203 is misleading. Is the analysis done using upregulated hub genes as mentioned in line 113-114? If so, does part of this result overlap with Figure 3C? The authors need to clarify this portion.
Minor Comments to improve
1) Misrepresentation in Figure 1 where HMMR was not identified from Gene enrichment analysis instead it was identified from shared hub gene of GSE31210 and TCGA. Arrow should be drawn from Identification of candidate hub gene to core gene identification.
2) X-axis of Figure 6 mentions time(years) whereas X-axis of Figure 7 mentions time(year). Following a uniform labeling is required.

Reviewer 3 ·

Basic reporting

The manuscript is presented in clear language with proper context and background. The figures are relevant, with a few minor mistakes in figure labeling that has been pointed out below. The authors address a relevant question that can potentially take the field forward with the identification of a gene that can be used as a prognostic marker for lung adenocarcinoma.

Experimental design

In the work provided, the authors analyzed multiple gene expression databases and associated bioinformatic studies to determine a potential core gene that can be used as a prognostic marker for lung adenocarcinoma (LUAD). The work is rigorous, and reasonably well-designed analyses have been chosen to address the question in concern.

Validity of the findings

The data provided are robust and statistically sound. Conclusions are well stated and within the reach of speculations. However, on fewer occasions, the reasoning of certain selections are unclear and noted below.

Additional comments

I have a few comments to the author that needs addressing for strengthening the submission.

Figure 2 - Please add the GSE profile identifier in figure 2 a-d.

In Figure 3a – Ube2c, CEP55, CDCA3 showed a higher degree of connectivity compared to HMMR. Any comment on why any of those genes were not pursued further as a potential biomarker?

In Figure 3c, biological pathways correlation of corresponding DEGs to PL1 signaling in the cell cycle has been completely ignored in favor of the FOXM1 network, though the PLK signaling dependent pathways are overabundantly present in the data. Please explain the choice in the result section.
Similarly, for the biological processes, more importance has been given to DEG’s association with the cell cycle. The potential connection to signal transduction and cell-cell communication (though found to be associated with the DEGs), thereby influencing various cancer pathways, has been overlooked. Please explain the particular selection process in the result section.

Figure 5I – The data shows the graph for the relative expression of KIF11. If this is the quantification of the western blot provided, mislabeling?

Line 171 – mislabeling Figure 5 ‘H and I’ as ‘E and F’?

Figure 6K – For the random-effects model, GSE 31210, 42127, and 684465 were used, although two other GSE databases were available for testing but were not used. Strongly recommend adding more data set here to increase confidence. GSE68465 showed a Hazard ratio of 1.0 as well. Also, please add the GSE database 42127 in the material and methods section; missing from the list of databases provided there.

In Figure 7B, the expression of HMMR above the age of 65 significantly impacted the survival rate, as shown in KM graph. Any comment or discussion on why that could be the result?

In the discussion section, the authors claimed to have provided robust evidence for the therapeutic role of HMMR in LUAD. All the data provided suggest a prognostic role of HMMR in LUAD. Please edit the sentence accordingly.

---

## Round 0.2 · accepted · Accept

All Reviewers agreed that the manuscript addresses all their concerns.

Reviewer 1 ·

Basic reporting

No Comment

Experimental design

No Comment

Validity of the findings

No Comment

Additional comments

The authors have addressed all my concerns, and the paper is now suitable for publication.

·

Basic reporting

Zhaodong Li et al has done an integrated bioinformatics analysis to identify HMMR as a prognostic biomarker for patients with lung adenocarcinoma. This study is well written with extensive literature references/relevant background with professional English.

Experimental design

Experimental design has been clearly documented with sufficient detail and information to replicate.

Validity of the findings

The authors have extensively modified the manuscript and most of the previous concerns have been addressed very clearly. The data is statistically sound and robust and conclusions are well supported by the data provided. I strongly believe that this manuscript can be found suitable for publication in PeerJ.

Reviewer 3 ·

Basic reporting

No Comments

Experimental design

No comments

Validity of the findings

No Comments

Additional comments

A minor comment, just editing changes.
Line 191 - Probably 'non-significant' will be a better fit.

The added changes have strengthened the manuscript.